# Assessment Study of ChatGPT-3.5’s Performance on the Final Polish Medical Examination: Accuracy in Answering 980 Questions

**DOI:** 10.3390/healthcare12161637

**Published:** 2024-08-16

**Authors:** Julia Siebielec, Michal Ordak, Agata Oskroba, Anna Dworakowska, Magdalena Bujalska-Zadrozny

**Affiliations:** Department of Pharmacotherapy and Pharmaceutical Care, Faculty of Pharmacy, Medical University of Warsaw, 02-091 Warsaw, Poland; s086967@student.wum.edu.pl (J.S.); agata.oskroba@wum.edu.pl (A.O.); anna.dworakowska@wum.edu.pl (A.D.); magdalena.bujalska@wum.edu.pl (M.B.-Z.)

**Keywords:** artificial intelligence, ChatGPT, medical exam

## Abstract

Background/Objectives: The use of artificial intelligence (AI) in education is dynamically growing, and models such as ChatGPT show potential in enhancing medical education. In Poland, to obtain a medical diploma, candidates must pass the Medical Final Examination, which consists of 200 questions with one correct answer per question, is administered in Polish, and assesses students’ comprehensive medical knowledge and readiness for clinical practice. The aim of this study was to determine how ChatGPT-3.5 handles questions included in this exam. Methods: This study considered 980 questions from five examination sessions of the Medical Final Examination conducted by the Medical Examination Center in the years 2022–2024. The analysis included the field of medicine, the difficulty index of the questions, and their type, namely theoretical versus case-study questions. Results: The average correct answer rate achieved by ChatGPT for the five examination sessions hovered around 60% and was lower (*p* < 0.001) than the average score achieved by the examinees. The lowest percentage of correct answers was in hematology (42.1%), while the highest was in endocrinology (78.6%). The difficulty index of the questions showed a statistically significant correlation with the correctness of the answers (*p* = 0.04). Questions for which ChatGPT-3.5 provided incorrect answers had a lower (*p* < 0.001) percentage of correct responses. The type of questions analyzed did not significantly affect the correctness of the answers (*p* = 0.46). Conclusions: This study indicates that ChatGPT-3.5 can be an effective tool for assisting in passing the final medical exam, but the results should be interpreted cautiously. It is recommended to further verify the correctness of the answers using various AI tools.

## 1. Introduction

The history of artificial intelligence (AI) dates back to the 1950s, when Alan Turing proposed the concept of machines capable of “thinking” like humans. As technology advanced, groundbreaking innovations such as deep learning emerged, enabling sophisticated data analyses. In recent years, the development of generative and multimodal AI has significantly impacted various fields, including medicine. Generative AI, exemplified by language models like GPT (Generative Pre-trained Transformer), and multimodal AI, which integrates different types of data (e.g., text and images), facilitate more comprehensive analyses and a better understanding of context [1,2,3]. To enhance the impact of this paper, a comparative analysis of ChatGPT-3.5’s performance with other AI models or tools used in similar educational contexts should be included. The literature review should be expanded to incorporate a detailed examination of studies on medical AI. Notably, several challenges persist in integrating vision-based AI systems into clinical practice. These challenges include the need for substantial computational resources, labor-intensive data annotation, domain shifting, and issues related to explainability. This study also addresses the often-overlooked challenge of data collection within the framework of contemporary data privacy laws. This research highlights that, while advancements focus on developing newer, larger models with quantitative metrics, there is a crucial need for ensuring that AI systems are trustworthy, ethical, and transparent. This perspective is well articulated in the work of Siriborvornratanakul [4]. Artificial intelligence tools, such as ChatGPT-3.5, are increasingly integrated into medical education and exam preparation. These tools facilitate personalized learning by offering tailored practice questions and adaptive feedback based on individual performance. AI-powered platforms can simulate clinical scenarios, enabling students to practice diagnostic and decision-making skills in a controlled environment. Additionally, AI is employed to generate practice exams that mirror the format and difficulty of actual medical tests, enhancing preparation efficiency. These platforms often use adaptive algorithms to adjust question difficulty, providing a dynamic and customized study experience. However, the effectiveness of these tools depends on the quality of data and algorithms, and ethical considerations must be addressed to avoid over-reliance. Overall, AI tools are becoming valuable assets in medical education, offering novel methods to support learning and exam preparation [5].

In recent years, there has been an increasing number of articles in the literature evaluating the accuracy of ChatGPT’s responses to questions appearing on medical exams. Conducting studies on the application of these advanced AI technologies in specific fields, such as medical examinations, is crucial for assessing their practical utility and impact, thereby underscoring the relevance and innovation of this research. The first article describing ChatGPT’s achievements on exams taken by medical students was published in the PubMed database in February 2023 [6]. Based on the published studies, it is challenging to definitively assess ChatGPT’s competence in passing medical exams. The current research presents mixed results, with some studies indicating that ChatGPT performs well on certain medical knowledge tests, while others highlight significant gaps in its understanding. Furthermore, the rapidly evolving nature of medical science means that ChatGPT’s knowledge base, which relies on pre-existing data, may not always be up to date with the latest medical advancements [7]. Based on the summary provided, Levin et al. conducted a meta-analysis of studies on ChatGPT’s performance in medical multiple-choice examinations. They searched the PubMed, Scopus, and Web of Science databases for relevant articles up to 2 June 2023, including peer-reviewed articles that assessed ChatGPT version 3.5. The exclusion criteria were non-English language performance and studies not utilizing multiple-choice questions. Two reviewers independently performed the review stages, resolving disagreements with a third reviewer. The analysis included 19 articles, showing that ChatGPT’s performance varied widely across different medical fields, and it established an average performance of 61.1% [8]. Sumbal et al. conducted a systematic review to assess the academic potential of ChatGPT-3.5 in medical exams, following PRISMA guidelines and searching databases such as PubMed/MEDLINE, Google Scholar, and Cochrane. They reviewed 12 articles that evaluated ChatGPT-3.5’s performance. The results indicated that ChatGPT performed well in four tests, average in four, and poorly in four, with performance directly related to the difficulty level of the questions. ChatGPT demonstrated strong explanation, reasoning, memory, and accuracy but struggled with image-based questions and lacked insight and critical thinking. That study concluded that, while ChatGPT-3.5 performed satisfactorily as an examinee, further research was needed to fully explore its potential in medical education [9]. Differences in the results may be due to various factors that differentiate the exams, such as the specific medical field or the particular skill being tested in applying theoretical knowledge. The volume of exams used in the tests varied, starting from about a hundred [7,10,11,12], moving on to several hundred [12,13,14,15,16], and ending with individual studies with around a thousand questions [17,18,19]. In the case of Poland, Wojcik et al. evaluated OpenAI’s ChatGPT-4 model in the context of a Polish medical specialization licensing exam (PES) conducted from 28–30 June 2023. Their study aimed to assess ChatGPT-4’s ability to answer medical questions and explore its potential impact on medical education and practice. The results showed that ChatGPT-4 achieved a 67.1% correct response rate, accurately answering 80 out of 120 questions posed during the exam. This performance highlights significant advancements in natural language processing relevant to medical education. The researchers concluded that, although ChatGPT can be a valuable tool in medical education, it cannot fully replace human expertise and knowledge due to its inherent limitations. Their findings underscore the evolving role of AI, exemplified by ChatGPT, in enhancing educational methodologies within the medical field [7]. Therefore, it seems reasonable to evaluate the capabilities of artificial intelligence on the primary and mandatory final exam for medical studies in Poland, namely the Medical Final Examination. For this exam, candidates undertake a test consisting of 200 single-choice closed questions from various fields of medicine. The exam is conducted by the Medical Examination Center. The questions for the Medical Final Examination in Poland are composed by a team of medical experts appointed by the relevant institutions overseeing medical exams. The exam is considered passed if the correctness of the answers is at least 56%. The analysis conducted for this article examines various fields of medicine where ChatGPT-3.5 has not been previously tested, including pharmacology. Additionally, it includes both knowledge-based and case study-type questions. The pool of questions used in the analysis totals 980, making this article one of the few to include such a large number of questions. Furthermore, the analysis considered the difficulty index provided by the Medical Examination Center for each question, ranging from 0 for extremely difficult tasks to 1 for extremely easy tasks. The lower its value, the more difficult the question. The test’s difficulty index was calculated as the average value of the difficulty indices of the individual tasks. Additionally, the correctness of the answers provided by the AI tool was compared to the average correctness of the answers provided by the examinees. While ChatGPT-4 has shown significant advancements, this study focused on ChatGPT-3.5 due to its availability. The choice of ChatGPT-3.5 allows for an evaluation of how earlier models perform in a complex medical examination context. Additionally, analyzing the performance of ChatGPT-3.5 provides a comparative perspective that can be valuable for understanding the evolution of AI models. This approach ensures that findings are relevant to both current and past versions of the technology. By examining ChatGPT-3.5, we aim to contribute to a broader understanding of AI capabilities and their implications in medical education. ChatGPT-3.5 shows a notable ability to understand and generate Polish text. The model generally performs well with standard Polish language usage but may face challenges with complex or specialized medical terminology. Given these capabilities, this study examines how effectively ChatGPT-3.5 processes Polish language in the context of the Polish Medical Final Examination. The analysis considered both the strengths and limitations of the model in handling medical language tasks in Polish. Despite advancements in evaluating AI models like ChatGPT on medical examinations, a knowledge gap exists in understanding the performance of earlier versions, such as ChatGPT-3.5, on comprehensive medical tests conducted in specific languages, such as Polish. In summary, this article analyzes the impact of factors such as the field of medicine, question difficulty, and the average number of correct answers provided by examinees for questions in each of the five exam sessions, each consisting of 200 questions.

## 2. Materials and Methods

### 2.1. The Accuracy of Responses Provided by ChatGPT

The analysis presented in this study focused on the correct responses provided by ChatGPT-3.5. ChatGPT-3.5 was selected for this research primarily due to its free and easy accessibility, which allows for widespread use without financial barriers. It is crucial to clarify that this study specifically utilized ChatGPT-3.5 and not the more advanced versions such as ChatGPT-4, ChatGPT-4o, or ChatGPT-4omini. These newer versions may offer enhanced accuracy and performance due to improvements in their underlying architecture and training data. By explicitly stating the version of ChatGPT used, we aim to provide a clear context for our findings and ensure that the results are interpreted with the appropriate consideration of the model’s capabilities. The analysis of correct responses provided by ChatGPT-3.5 to questions from the Medical Final Examination was conducted in April and May 2024. The source of the analyzed questions was the open database of the Medical Examination Center (CEM) in Łódź [20]. This study included questions from the Spring 2024, Fall 2023, Spring 2023, Fall 2022, and Spring 2022 sessions. Each session contained 200 questions, so a total of 1000 questions were tested in this study. However, the analysis included a total of 980 questions as 20 questions were withdrawn by the Medical Examination Center. Information about the difficulty index of the questions and which of the assigned answers for a specific question were correct was also obtained from the CEM. The database included two types of questions: theoretical and case study. Each question session was preceded by the prompt, “I will now ask you a few questions; choose only one answer, i.e., A, B, C, D, or E”. Questions from each session were asked in a separate chat. In cases of a system crash or an inability to answer questions, the remaining questions from that session were asked in a newly created chat. Although ChatGPT-3.5 was instructed to choose a single correct answer for each question, there were instances where the model provided multiple or ambiguous responses. In such cases, the evaluation was based on the single correct answer assigned to each question by the Medical Examination Center. This approach applied to both theoretical questions and case study descriptions. If ChatGPT-3.5’s output did not directly match the correct answer provided by the Medical Examination Center, the response was considered incorrect. The evaluation process ensured that all responses were assessed in alignment with the definitive answers established by the Medical Examination Center, maintaining consistency and accuracy in the assessment of the model’s performance. The correctness of the responses was assessed both overall and divided by medical field. The obtained results were compared to the difficulty index of the analyzed questions. Additionally, the percentage of correct answers was compared to the average score achieved by examinees in Poland during each of the 5 mentioned time periods. The average scores achieved by examinees were gathered from the Medical Examination Center records, which provided aggregated performance data for each exam session. The data included the overall average scores. The difficulty index of each question, provided by the Medical Examination Center, was used to categorize the questions based on how challenging they were, with higher indices indicating easier questions. The difficulty index of a task was calculated using the following formula: IDI = (Ns + Ni)/2*n*. In this formula, *n* represents the number of examinees in each extreme group (the top 27% and bottom 27% of scorers), Ns denotes the number of correct responses in the highest-scoring group, and Ni indicates the number of correct responses in the lowest-scoring group. This index ranges from 0 (extremely difficult tasks) to 1 (extremely easy tasks), and it is distinct from the percentage of correct responses as it excludes those given by examinees with average test scores. The overall difficulty index of the test is the mean value of the difficulty indices of individual tasks. To establish the ground truths for the correctness of responses, we relied on aggregated performance data provided by the Medical Examination Center, which is a well-recognized and reliable source for such data. This information was integrated into the analysis to evaluate ChatGPT-3.5′s performance relative to the difficulty of the questions. By comparing the percentage of correct answers provided by ChatGPT-3.5 with these average scores and difficulty indices, we ensured a nuanced assessment of the model’s performance across various dimensions of the Medical Final Examination.

### 2.2. Statistical Analysis

Statistical analysis was performed using the SPSS25 statistical package (IBM SPSS Statistics, IBM Corp. (Armonk, NY, USA)). To determine whether there is a statistically significant relationship between variables measured on a nominal scale, the chi-square test was applied. The effect size was measured using the phi coefficient. The eta coefficient was used to analyze the relationship between a nominal variable and a quantitative variable, which can take values ranging from 0 to 1, where values close to 1 indicate a strong correlation and values closer to 0 indicate a weaker correlation. The Mann–Whitney U test was employed to check for statistically significant differences between the two groups of questions divided based on their difficulty index. A *p* value of < 0.05 was taken as the statistically significant level.

## 3. Results

### 3.1. The Accuracy of the Provided Answers

The accuracy of the correct answers provided by the artificial intelligence tool hovered around 60% (Figure 1). This was lower compared to the percentage of correct answers given by exam takers (*p* < 0.001; ø = 0.24).

The score achieved by ChatGPT in each examination session was slightly above the required minimum of 56% but was lower than the average score achieved by examinees in the five time periods (Table 1).

When dividing the questions into individual medical disciplines (*n* = 21), the percentage of correct responses ranged from 42.1 to 78.6. The lowest percentage pertained to hematology, while the highest was observed in endocrinology. Significantly more correct responses were provided for endocrinology, oncology, psychiatry, public health, and emergency medicine. For the remaining questions, a similar percentage of correct and incorrect responses was noted. Additionally, pharmacology was included in the table, where the percentage of correct responses closely approximated that of the other disciplines; specifically, just under 64% (Table 2, *n* = 980). Among the mentioned fields, medical jurisprudence involves the study and application of legal principles to medical practice, including the adjudication of medical-related legal cases and the interpretation of laws in medical contexts. Medical law encompasses the broader legal framework governing healthcare, such as patient rights, healthcare policies, and professional responsibilities. Together, these areas ensure that medical practices adhere to legal standards and protect the rights of both patients and healthcare providers.

### 3.2. Difficulty Index of the Questions

The difficulty index of the questions showed a statistically significant relationship with the correctness of responses (eta = 0.69; *p* = 0.04). This indicates that the difficulty index moderately influences whether an answer is correct. In other words, more difficult questions tend to result in a higher number of incorrect responses. Confirmation of these findings is evidenced by a statistically significant difference between incorrect versus correct responses to questions in terms of their difficulty index (*p* < 0.001). A lower difficulty index corresponds to more difficult questions. Incorrect responses were more frequent for questions with higher difficulty (Figure 2).

Questions with a difficulty index of up to 0.5 had only 37.1% correct responses. Conversely, questions with a difficulty index greater than 0.5 had a higher correct response rate of 63.9% (Figure 3). This difference is statistically significant (*p* < 0.001; effect size = 0.14).

### 3.3. The Type of Questions under Analysis

Among the analyzed questions, 101 were case studies, while the remainder were theoretical. There were no differences found in the percentage of correct responses provided by the artificial intelligence tool for theoretical questions versus case studies (*p* = 0.46) (Figure 4).

## 4. Discussion

In recent years, artificial intelligence (AI) has achieved significant milestones, gaining widespread recognition across various fields. Chat Generative Pre-Trained Transformer (ChatGPT), a language model developed by OpenAI, has gained considerable popularity since its public debut in 2022. Alongside numerous other chatbots, ChatGPT has sparked interest both among the general public and in academic circles. The ability to provide immediate responses to questions presents an opportunity to enhance service quality for doctors, patients, and healthcare professionals. Gilson et al. observed that ChatGPT is capable of achieving results comparable to those of a third-year medical student in the United States [21]. The aforementioned study revealed that ChatGPT-3.5 achieved an average correct answer rate of approximately 60% on the Final Medical Examination required for full medical licensure in Poland. The analysis indicated that the accuracy of ChatGPT-3.5’s responses is significantly influenced by the difficulty of the questions, with more challenging questions correlating with a lower rate of correct answers. Conversely, the type of questions did not substantially affect the correctness of the responses. These findings suggest that, while ChatGPT-3.5 has potential as a tool for assisting in medical exam preparation, its responses should be interpreted with caution. Further verification of AI-generated answers using multiple tools is advisable to ensure accuracy and reliability in high-stake assessments.

The results obtained indicate that the accuracy of responses provided by the AI tool varied around 60%. These findings are consistent with those reported in a systematic review by Levin et al., where ChatGPT’s performance ranged from 40% to 100%, with an average of 60% [8]. However, this percentage was lower than the results observed in the five sessions of the medical licensure examination in this study. It should be noted that most current medical exams primarily assess the ability to recall information from memory [7]. Stengel et al. investigated the accuracy of AI in the written portion of the EANS board exam using 86 representative single-best-answer questions. Their study classified the questions as text-based or image-based and tested them with ChatGPT-3.5, Bing, and Bard. Bard achieved the highest accuracy with 62% correct answers overall and 69% when excluding image-based questions, outperforming human participants. All AI models exceled at theory-based questions but failed to answer any image-based questions correctly. Their study concluded that AI can pass the written EANS board exam and perform comparably to or better than human participants, raising important ethical and practical considerations for the future design of the exam [10]. Depending on the medical discipline, the results of this study ranged from 42.1% for hematology to 78.6% for endocrinology. An article published in October 2023 in the Journal of Diabetes Science and Technology focused on the multiple-choice question (MCQ) examination format. ChatGPT’s accuracy in endocrinology-related questions was reported at 58% [22]. Psychiatry was among the areas where ChatGPT achieved a notably high percentage of correct answers. D’Souza et al. conducted an experimental study to evaluate ChatGPT-3.5’s performance in Psychiatry using 100 clinical case vignettes. Expert psychiatry faculties assessed ChatGPT-3.5’s responses, categorizing them into 10 classifications including management strategies, diagnosis, and ethical reasoning. The results showed ChatGPT-3.5 received “Grade A” ratings in 61 cases, “Grade B” in 31, and “Grade C” in 8, indicating strong performance with minimal errors. The AI excelled particularly in generating management strategies and diagnoses across various psychiatric conditions [23]. Published articles thus far have indicated significant variations in ChatGPT’s performance across different medical disciplines and levels of question specificity, with percentage differences between disciplines ranging from 10% to 15% [24]. According to data published in JAMA Ophthalmology and Ophthalmol Science, ChatGPT performs well in general medicine topics but struggles with more specialized issues [13,24]. Differences also exist between various medical specialties; Flores-Cohalia et al. found that ChatGPT performed 5–10% worse in obstetrics and gynecology compared to emergency medicine or pediatrics [25]. Given these significant differences between disciplines, testing across a wide range of fields is crucial. Their article also analyzed the accuracy of responses in pharmacology-related questions. The article published in the Journal of Pharmacy Practice discussed the potential of AI-powered generative language models, specifically ChatGPT, in assisting healthcare professionals, including pharmacists, with their daily tasks. Their study aimed to evaluate ChatGPT’s effectiveness in answering various pharmacy-based questions. Findings from the aforementioned study showed that ChatGPT achieved an overall score of 47.73% correct responses across 32 questions categorized into drug information, patient cases, calculations, and drug knowledge. It performed well in pharmacy calculations (100%) and knowledge of the top-200 drugs (80%) but had lower scores in drug information with enhanced prompts (33%) and patient cases (20%). Their study concluded that, while ChatGPT shows potential in certain areas of pharmacy practice, its performance varies significantly depending on the complexity of the questions posed [26].

Recent studies have reported findings similar to those observed in the manuscript, particularly in relation to Polish examinations and the performance of ChatGPT. Kufel et al. investigated ChatGPT’s performance on the Polish national specialty examination (PES) in radiology and imaging diagnostics. Their results indicated that ChatGPT did not meet the PES pass rate threshold of 52%, although it performed reasonably well in certain categories. This is consistent with the manuscript’s findings, which highlight that ChatGPT’s performance can vary and may not always meet the required standards [27]. Nicikowski et al. analyzed ChatGPT-3.5 and ChatGPT-4.0 on nephrology-specialty questions from the Polish Medical Examination Center. They reported that ChatGPT-4.0 achieved a 69.84% accuracy rate, significantly outperforming ChatGPT-3.5 (45.70%) and surpassing the human average (85.73%). This finding aligns with the manuscript’s results, where newer versions of ChatGPT showed improved performance in specific areas, though some challenges remained [28]. Lewandowski et al. assessed ChatGPT-3.5 and ChatGPT-4 in dermatology exams, with ChatGPT-4 exceeding the 60% pass rate in all tests and achieving high accuracy rates of 80% and 70% for the English and Polish versions, respectively. This significant improvement over ChatGPT-3.5 corresponds with this manuscript’s observations, indicating that advanced versions of ChatGPT perform better but still exhibit variability across different disciplines [29]. These findings from various Polish studies underscore that while ChatGPT has potential in medical examinations, its performance is not uniformly reliable and may not always meet the desired accuracy levels, reflecting the results discussed in the manuscript.

Incorrect responses from the AI tool may result from its general nature as a language model, which lacks specialization in medicine and therefore limits its ability to provide accurate answers in this domain. To improve its performance, more precise adaptation through fine tuning on medical datasets and the use of prompt-tuning techniques are necessary. The absence of fine-tuning options for GPT-3.5 and GPT-4 models currently available from OpenAI may further contribute to imperfections in their medical context responses [30,31]. Difficult questions, as indicated by the difficulty index used in this study, were characterized by a lower percentage of correct responses. This was corroborated by the research published in June 2024 in Scientific Reports, which examined ChatGPT’s performance on a selection of USMLE Step 1 practice questions (United States Medical Licensing Examination). Their study found that ChatGPT achieved an overall accuracy rate of 55.8% across 2377 text-based questions from the Amboss question bank. It showed varying success across different question categories, performing well in serology-related queries but struggling with ECG-related content. Despite these challenges, ChatGPT demonstrated consistent performance across different levels of question difficulty. These findings highlight the potential of ChatGPT in medical education while suggesting areas where further improvements are needed [32]. Sumbal et al. pointed out that ChatGPT performs less effectively in questions requiring critical thinking [9]. In the near future, it is unlikely that AI will replace healthcare professionals. It is important to remember that advanced algorithms and AI-supported technologies currently lack the capability to autonomously diagnose and treat diseases without active human involvement. The practical implications of these findings for medical educators and students are significant. For educators, the results highlight the importance of integrating AI tools like ChatGPT into the curriculum as supplementary resources to enhance learning and exam preparation. Understanding the variability in AI performance across different medical disciplines can help educators tailor their teaching strategies to address the gaps identified by AI, thereby improving overall educational outcomes. For students, these insights emphasize the necessity of using AI-generated responses as a supportive tool rather than as a definitive source of information. This approach encourages critical thinking and cross-referencing with reliable medical sources, ultimately fostering a more comprehensive understanding of medical knowledge. Furthermore, the findings suggest that while AI tools can aid in studying, they should not replace traditional learning methods and human judgment, especially in high-stake examinations and clinical decision making. Therefore, the adoption of AI in medical education should be conducted with caution, ensuring continuous assessment and improvement of these technologies to align with educational standards and practices [33].

## 5. Conclusions

This study demonstrates that ChatGPT-3.5 achieved an average accuracy of about 60% on the Final Medical Examination, which is lower than that of human examinees. The model’s performance was significantly affected by question difficulty, with more challenging questions leading to lower accuracy. Although ChatGPT-3.5 has potential as a tool for medical exam preparation, its responses should be interpreted cautiously. Further research is needed to validate the accuracy of AI models and assess their effectiveness across different types of exam questions.

## 6. Limitations

One notable limitation of this study lies in its failure to compare outcomes with alternative AI models or conversational systems. To address this gap, future research endeavors should explore the use of advanced AI tools such as ChatGPT-4, which has been acknowledged for its capability to deliver more refined and accurate outputs. Despite this recommendation, the current study primarily employed ChatGPT-3.5 due to its cost effectiveness and widespread accessibility, factors that are often crucial in research settings constrained by budgetary considerations. Moreover, it is essential to recognize that ChatGPT-3.5’s training data predate 2021, potentially hindering its ability to incorporate the latest advancements and nuances in the field of artificial intelligence. This temporal limitation underscores the necessity for researchers to remain vigilant of the evolving landscape of AI technologies and their implications for research outcomes. Furthermore, the suggestion to repeat identical inquiries multiple times with intervals is grounded in prior research findings indicating the potential for varied responses. This methodological approach could unveil insights into the reliability and consistency of AI-generated responses, thus enriching the validity of findings in studies involving conversational AI systems like ChatGPT-3.5. In summary, while ChatGPT-3.5 serves as a practical choice for this study, future investigations should consider leveraging more advanced AI models like ChatGPT-4 to enhance the robustness and comprehensiveness of their research outcomes. Additionally, vigilance regarding the temporal constraints of AI training data and methodological considerations such as repeated questioning could further refine the quality and applicability of research findings in the realm of artificial intelligence. A significant limitation of this study is its reliance on a limited set of Polish medical exam questions, which may not fully represent the diversity of content encountered in broader medical licensure assessments. If these exams were available as open source, it would be pertinent to explore whether ChatGPT-3.5 might have been exposed to similar material during its training phase. This prior exposure could potentially influence the model’s performance on such questions. Future research should address these limitations by incorporating a wider array of exam content and examining how prior training data might impact an AI’s responses.

## Figures and Tables

**Figure 1 healthcare-12-01637-f001:**
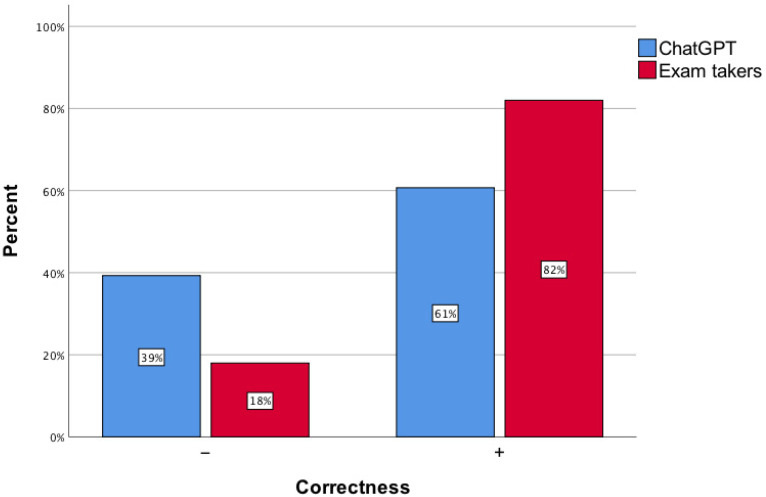
The accuracy of responses to the Medical Final Examination questions by examinees and ChatGPT.

**Figure 2 healthcare-12-01637-f002:**
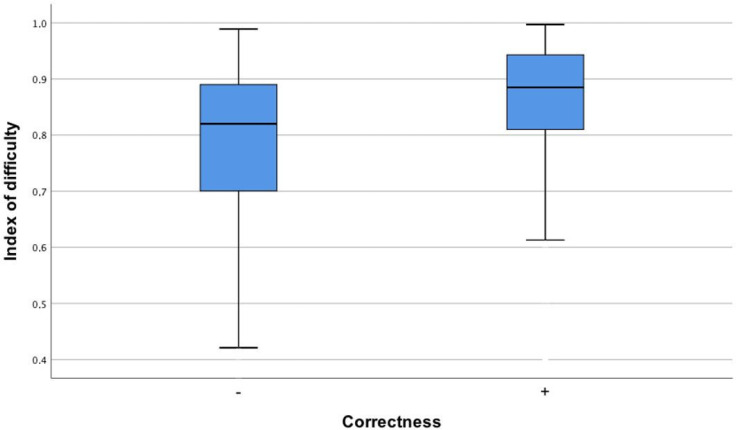
The difficulty index of questions in the Final Medical Examination for questions with correct and incorrect responses provided by ChatGPT.

**Figure 3 healthcare-12-01637-f003:**
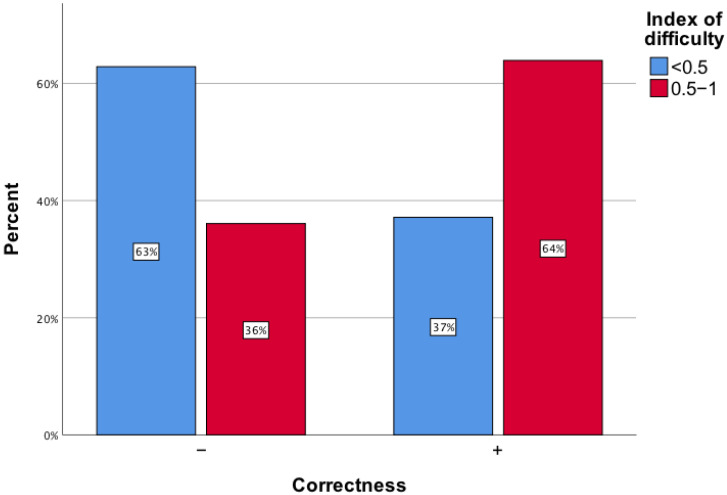
The accuracy of correct responses provided by ChatGPT on questions from the Final Medical Examination categorized by difficulty index values of up to 0.5 and between 0.5 and 1.

**Figure 4 healthcare-12-01637-f004:**
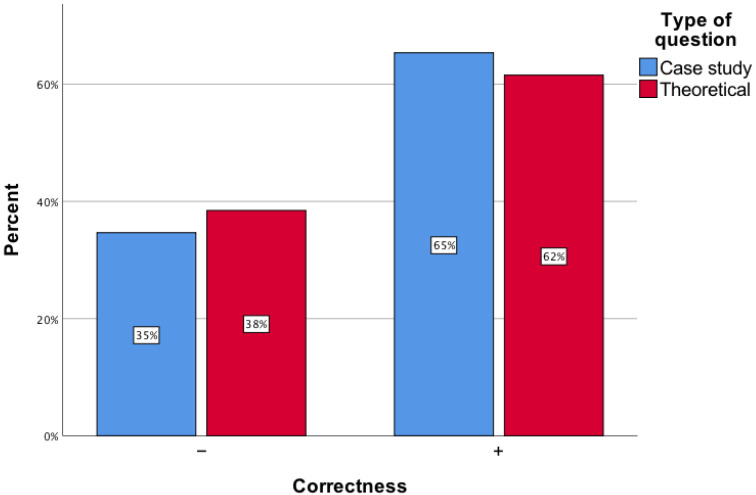
The accuracy of responses provided by ChatGPT on theoretical questions versus case study questions.

**Table 1 healthcare-12-01637-t001:** The number and percentage of correct answers provided by ChatGPT, and the average score achieved by those taking the Medical Final Examination over five time periods.

Date	*n*	%	The Average Score Obtained Nationally		*p*-Value * and Effect Size (Phi Coefficient)
			*n*	%	
Spring 2022	119	59.5	165	82.7	*p* < 0.001; ø = 0.25
Autumn 2022	115	57.5	166	82.8	*p* < 0.001; ø = 0.28
Spring 2023	127	63.5	162	81	*p* < 0.001; ø = 0.2
Autumn 2023	124	62	163	81.7	*p* < 0.001; ø = 0.22
Spring 2024	122	61	164	81.8	*p* < 0.001; ø = 0.23

* chi-squared.

**Table 2 healthcare-12-01637-t002:** Percentage of correct responses provided by ChatGPT on questions from the Final Medical Examination when divided by medical disciplines.

Discipline	% of Correct Answers (*n* = 980)	*p*-Value *
Endocrinology	78.6	*p* = 0.002
Pulmonology	64.3	*p* = 0.29
Cardiology	57.8	*p* = 0.3
Gastroenterology	54.3	*p* = 0.56
Oncology	68.1	*p* < 0.001
Nephrology	60	*p* = 0.27
Internal medicine	55.7	*p* = 0.31
Pediatrics	58.7	*p* = 0.17
Surgery	53.8	*p* = 0.78
Gynecology	57.4	*p* = 0.28
Obstetrics	48	*p* = 0.84
Hematology	42.1	*p* = 0.49
Neonatology	52.9	*p* = 0.21
Diabetology	66.7	*p* = 0.25
Psychiatry	70.3	*p* = 0.001
Family medicine	62.9	*p* = 0.13
Medical jurisprudence	48.6	*p* = 0.87
Public health	75.6	*p* = 0.001
Emergency medicine	64.2	*p* = 0.006
Medical law	64.3	*p* = 0.06
Pharmacology	63.8	*p* = 0.007
Other	58.2	*p* = 0.047

* chi-squared.

## Data Availability

The datasets used and analyzed during the current study are available from the corresponding author upon reasonable request.

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
