# Peer review of "Assessment Study of ChatGPT-3.5’s Performance on the Final Polish Medical Examination: Accuracy in Answering 980 Questions"

_healthcare, 2024, doi:10.3390/healthcare12161637_

Round 1

Reviewer 1 Report

Comments and Suggestions for Authors

General comments

=============

I appreciate the opportunity to review your article titled “Can ChatGPT-3.5 pass the Final Polish Medical Examination? Accuracy in answering 1000 questions.” While the topic is intriguing, the manuscript would greatly benefit from significant elaboration throughout to enhance its scholarly contribution.

Specific comments

=============

Major comments

---------------------

[Overall]

- Clarify the generation of ChatGPT as ChatGPT-3.5. This distinction is crucial since this study did not use more advanced versions like ChatGPT-4, ChatGPT-4o, or ChatGPT-4omini, which might offer higher accuracy.

[Title]

- The title should specify the study type, such as an experimental study, to provide clarity on the research design.

[Abstract]

- Clarify the language used for the final medical examination in Poland.

- Clarify the style of the medical final examination in Poland, whether it is composed of multiple-choice questions, written exams, or a combination of both.

[Introduction]

- Provide a broader context by initially discussing the application of AI in medicine before narrowing down to radiology. Additionally, include a detailed historical overview of AI, especially generative and multimodal AI, to underscore the study's relevance and innovation.

- The introduction mentions Polish accuracy for ChatGPT-4, but this study used ChatGPT-3.5. Please clarify this discrepancy and explain the rationale for focusing this study on the Polish medical exam using ChatGPT-3.5.

- Clarify the performance capabilities of ChatGPT-3.5 for understanding and processing Polish language.

- Highlight the knowledge gap addressed by this study before stating the study's aim.

[Methods]

- Clarify how the outputs from ChatGPT-3.5 were evaluated, especially in cases where the output did not choose a single answer.

- Explain how the average score achieved by examinees was gathered, including the difficulty index and the subfields of the exam.

[Results]

- Provide the number of included medical questions and their subfields. Additionally, include the number of medical disciplines covered in Table 2.

- Provide details on Medical Jurisprudence and Medical Law for international readers to understand these terms better.

[Discussion]

- Initially, discuss the primary outcomes of this study.

- Directly compare your findings with previous studies involving Polish examinations or ChatGPT-3.5, avoiding references to introduction sections without direct discussion.

- Discuss the limitation that this study employed a limited medical exam in Polish. If these exams were available as open-source, discuss the potential prior learning by the GPT model during its development.

[Conclusion]

- Clearly state the conclusions drawn from the study's findings. The current conclusion appears too broad and should be more tightly aligned with the study's results.

These comments are structured to help refine the manuscript significantly, aiming to provide clarity, enhance methodological transparency, and deepen the contextual background of the research study.

Comments on the Quality of English Language

Please refer the previous quality of English language.

Author Response

Dear Reviewer nr 1,

Comment 1: “I appreciate the opportunity to review your article titled “Can ChatGPT-3.5 pass the Final Polish Medical Examination? Accuracy in answering 1000 questions.” While the topic is intriguing, the manuscript would greatly benefit from significant elaboration throughout to enhance its scholarly contribution.”

Answer 1: Thank you for the valuable advice, based on which we have included an additional description in the methodology as you suggested.

Comment 2: “Revising the title to reflect on the study results (1st part) and study design (2nd part) is advisable.”

Answer 2: The title of the manuscript has been changed to indicate that it is an assessment. The title has been updated to reflect the number of questions as 980 instead of 1000, due to the exclusion of 20 questions by the Medical Examination Center.

Comment 3: “Clarify the language used for the final medical examination in Poland.”

Answer 3: In the abstract, the issue of the exam's language was addressed.

Comment 4: “Clarify the style of the medical final examination in Poland, whether it is composed of multiple-choice questions, written exams, or a combination of both?”

Answer 4: In the abstract, it is specified that these are questions with one correct answer per question.

Comment 5: “Provide a broader context by initially discussing the application of AI in medicine before narrowing down to radiology. Additionally, include a detailed historical overview of AI, especially generative and multimodal AI, to underscore the study's relevance and innovation.”

Answer 5: Thank you for the valuable feedback. The introduction of the manuscript has been expanded to include this aspect.

Comment 6: “The introduction mentions Polish accuracy for ChatGPT-4, but this study used ChatGPT-3.5. Please clarify this discrepancy and explain the rationale for focusing this study on the Polish medical exam using ChatGPT-3.5.”

Answer 6: Thank you for your feedback. At the end of the introduction, we have added a few sentences explaining the choice of this model.

Comment 7: “Clarify the performance capabilities of ChatGPT-3.5 for understanding and processing Polish language.”

Answer 7: This aspect has also been added to the introduction.

Comment 8: “Highlight the knowledge gap addressed by this study before stating the study's aim.”

Answer 8: Thank you for the additional feedback on the introduction. We have added this type of information.

Comment 9: “Clarify how the outputs from ChatGPT-3.5 were evaluated, especially in cases where the output did not choose a single answer.” 

Answer 9: This type of information has also been added to the methodology section.

Comment 10: “Explain how the average score achieved by examinees was gathered, including the difficulty index and the subfields of the exam.”

Answer 10: Such indicated information has been added to the manuscript.

Comment 11: “Provide the number of included medical questions and their subfields. Additionally, include the number of medical disciplines covered in Table 2.”

Answer 11: The number of tested questions and disciplines has been added.

Comment 12: “Provide details on Medical Jurisprudence and Medical Law for international readers to understand these terms better.”

Answer 12: Before Table 2, the details of these disciplines are outlined.

Comment 13: “Initially, discuss the primary outcomes of this study.”

Answer 13: At the beginning of the discussion, sentences were added to reflect the obtained results.

Comment 14: “Directly compare your findings with previous studies involving Polish examinations or ChatGPT-3.5, avoiding references to introduction sections without direct discussion.”

Answer 14: In accordance with the received advice, tasks related to Polish exams have been added to the discussion.

Comment 15: “Discuss the limitation that this study employed a limited medical exam in Polish. If these exams were available as open-source, discuss the potential prior learning by the GPT model during its development.”

Answer 15: Such a limitation has been added to the manuscript.

Comment 16: “Clearly state the conclusions drawn from the study's findings. The current conclusion appears too broad and should be more tightly aligned with the study's results.”

Answer 16: The conclusions have been revised.

Comment 17: “Please refer the previous quality of English language.”

Answer 17: Several sentences have been corrected for grammatical accuracy.

Reviewer 2 Report

Comments and Suggestions for Authors

The paper presents an analysis of ChatGPT-3.5's performance on 1000 questions from the Medical Final Examination conducted in Poland between 2022 and 2024. The study assesses the AI model's effectiveness across different medical fields, difficulty levels, and question types. The findings indicate that while ChatGPT-3.5 shows potential in aiding medical exam preparation, it does not outperform human examinees and varies significantly in accuracy across different medical specializations.

The paper is written well. The topic of the paper is technically sound, and the idea of the proposed method is interesting. However, the following issues should be addressed before making a final decision. First, while the study is focused on ChatGPT-3.5's performance, it would benefit from more context on how such AI tools are currently integrated into medical education and exam preparation. Second, the paper could enhance its impact by comparing ChatGPT-3.5's performance with other AI models or tools used in similar educational contexts. The literature review should be conducted more deeply. The papers on medical AI should be added and discussed, including “Advanced Artificial Intelligence Methods for Medical Applications. HCI (19) 2023: 329-340”. Third, although the paper reports statistical significance for various factors, it could provide more detailed explanations of the practical implications of these findings for medical educators and students. The difficulty index of questions in the Final Medical Examination for questions should be elaborated more clearly. How do the authors get the ground truths? Is it reliable enough? Moreover, the authors should explain how to get the accuracy of responses provided by ChatGPT on theoretical questions versus case study questions more clearly.

Overall, the paper provides a valuable contribution to the understanding of AI's role in medical education. Addressing the identified weaknesses will enhance its relevance and impact. I, thus, recommend accepting after suitable changes.

Comments on the Quality of English Language

The paper presents an analysis of ChatGPT-3.5's performance on 1000 questions from the Medical Final Examination conducted in Poland between 2022 and 2024. The study assesses the AI model's effectiveness across different medical fields, difficulty levels, and question types. The findings indicate that while ChatGPT-3.5 shows potential in aiding medical exam preparation, it does not outperform human examinees and varies significantly in accuracy across different medical specializations.

The paper is written well. The topic of the paper is technically sound, and the idea of the proposed method is interesting. However, the following issues should be addressed before making a final decision. First, while the study is focused on ChatGPT-3.5's performance, it would benefit from more context on how such AI tools are currently integrated into medical education and exam preparation. Second, the paper could enhance its impact by comparing ChatGPT-3.5's performance with other AI models or tools used in similar educational contexts. The literature review should be conducted more deeply. The papers on medical AI should be added and discussed, including “Advanced Artificial Intelligence Methods for Medical Applications. HCI (19) 2023: 329-340”. Third, although the paper reports statistical significance for various factors, it could provide more detailed explanations of the practical implications of these findings for medical educators and students. The difficulty index of questions in the Final Medical Examination for questions should be elaborated more clearly. How do the authors get the ground truths? Is it reliable enough? Moreover, the authors should explain how to get the accuracy of responses provided by ChatGPT on theoretical questions versus case study questions more clearly.

Overall, the paper provides a valuable contribution to the understanding of AI's role in medical education. Addressing the identified weaknesses will enhance its relevance and impact. I, thus, recommend accepting after suitable changes.

Author Response

Dear Reviewer nr 2,

Comment 1: “The paper presents an analysis of ChatGPT-3.5's performance on 1000 questions from the Medical Final Examination conducted in Poland between 2022 and 2024. The study assesses the AI model's effectiveness across different medical fields, difficulty levels, and question types. The findings indicate that while ChatGPT-3.5 shows potential in aiding medical exam preparation, it does not outperform human examinees and varies significantly in accuracy across different medical specializations. The paper is written well. The topic of the paper is technically sound, and the idea of the proposed method is interesting. However, the following issues should be addressed before making a final decision.”

Answer 1: Thank you for the positive feedback on our manuscript and the advice we received, which we have addressed. The title has been updated to reflect the number of questions as 980 instead of 1000 due to the exclusion of 20 questions by the Medical Examination Center, and the correction was made after receiving advice from the first reviewer.

Comment 2: “First, while the study is focused on ChatGPT-3.5's performance, it would benefit from more context on how such AI tools are currently integrated into medical education and exam preparation.”

Answer 2: Thank you for this valuable and important advice. The manuscript's introduction has been expanded to include this aspect.

Comment 3: “Second, the paper could enhance its impact by comparing ChatGPT-3.5's performance with other AI models or tools used in similar educational contexts. The literature review should be conducted more deeply. The papers on medical AI should be added and discussed, including “Advanced Artificial Intelligence Methods for Medical Applications. HCI (19) 2023: 329-340”

Answer 3: The indicated information, based on the provided reference, has been added to the manuscript.

Comment 4: “Third, although the paper reports statistical significance for various factors, it could provide more detailed explanations of the practical implications of these findings for medical educators and students. The difficulty index of questions in the Final Medical Examination for questions should be elaborated more clearly. How do the authors get the ground truths? Is it reliable enough? Moreover, the authors should explain how to get the accuracy of responses provided by ChatGPT on theoretical questions versus case study questions more clearly.”

Answer 4: According to the advice of the first reviewer, the methodology has been expanded with additional sentences regarding average scores and the criteria for determining whether an answer is correct or not (based on the correct answers provided by the Medical Examination Center). This applies to all questions, both theoretical and case-based. However, with your advice, an additional sentence has been included in the supplementary description indicating that this applies to both theoretical questions and case descriptions. Additionally, we expanded the description of the difficulty index, providing justification for its selection. In accordance with the subsequent advice, the implications mentioned by the reviewer have been included in the discussion.

Round 2

Reviewer 1 Report

Comments and Suggestions for Authors

General comments

=============

I appreciate the opportunity to review your article titled “Can ChatGPT-3.5 pass the Final Polish Medical Examination? Accuracy in answering 1000 questions.” The topic is intriguing. Almost all responses were reasonable.